# Diagnostic Accuracy of Oral Cancer and Suspicious Malignant Mucosal Changes among Future Dentists

**DOI:** 10.3390/healthcare9030263

**Published:** 2021-03-02

**Authors:** Kamis Gaballah, Asmaa Faden, Fatima Jassem Fakih, Anfal Yousuf Alsaadi, Nadeen Faiz Noshi, Omar Kujan

**Affiliations:** 1Department of Clinical Sciences, College of Dental Sciences, Ajman University, P.O. Box 346 Ajman, United Arab Emirates; kamisomfs@yahoo.co.uk (K.G.); f.j.f_world@hotmail.com (F.J.F.); ko5011@outlook.com (A.Y.A.); naden_noshi@yahoo.com (N.F.N.); 2Department of Oral Medicine and Diagnostic Science, College of Dentistry, King Saud University, Riyadh 4545, Saudi Arabia; afaden@ksu.edu.sa; 3Oral Diagnostic and Surgical Sciences, UWA Dental School, The University of Western Australia, Nedlands, WA 6009, Australia

**Keywords:** diagnostic accuracy, dental student, dentists, oral cancer, oral potentially malignant disorders

## Abstract

This study aimed to assess the ability of dental students and recent graduates to detect and recognize mucosal changes that are suggestive of oral cancer and potentially malignant disorders. In this cross-sectional study, a questionnaire was administered to dental students and recent graduates of Ajman University (*n* = 132). Completed questionnaires were received from 84 (63.6%) females and 48 (36.4%) males which included fifth-year students (*n* = 80), interns (*n* = 39), and dental practitioners (*n* = 13). This questionnaire was designed to assess the respondent’s ability to detect and recognize different types of oral lesions, as well as their knowledge of oral cancer appearance and malignancy potential. The overall accuracy of diagnosis was 46%. The participants correctly identified normal variations, benign tumors, malignant tumors, and premalignant lesions at rates of 60.3%, 31.0%, 55.7%, and 33.4%, respectively. There was no significant difference between the two genders in their ability to recognize and detect correct answers (females, 48.3%; males, 47.2%). According to education level, interns provided the highest percentage of correct answers (52.5%), followed by newly dental practitioners (51.9%) and fifth-year students (44.1%). Conclusion: The respondents of this survey did not exhibit a satisfactory diagnostic capability in recognizing mucosal changes consistent with the clinical presentation of oral cancer. Thus, a need exists for improved and updated educational methods for undergraduate students regarding oral cancer and potentially malignant disorders. Meanwhile, practitioners should look for oral abnormalities to provide better diagnosis and management. Practitioners should also stay up to date on the oral malignancy topic by attending workshops and clinicopathological conferences.

## 1. Introduction

Oral cancer is a deleterious disease which can manifest in any part of the oral cavity or lips. The mortality rate for oral cancer is high since it is often diagnosed in its later stages, despite the fact that it can be early detected by standard oral examination [1]. The incidence of malignant oral neoplasms varies worldwide, largely due to differences in the distribution of risk factors and possible etiologies [1]. It is estimated that there are approximately 350,000 new cases of oral cancers each year, and more than 330,000 deaths [2]. Tobacco use, including smokeless tobacco, and excessive alcohol consumption, are estimated to account for approximately 90% of oral cancers [1].

Oral squamous cell carcinomas (OSCCs) are frequently preceded by white or red colored mucosal changes known as leukoplakia or erythroplakia [3]. A lesion that is both red and white in color is referred to as erythroleukoplakia [4]. In the latter mixed lesion, there is an increased risk of invasive carcinoma cells being present, leading to their classification as “potentially malignant disorders” [3]. The majority of leukoplakias will not proceed to cancer. However, their potential for malignant transformation is well documented [4]. Meanwhile, erythroplakias will show evidence of high-grade epithelial dysplasia, carcinoma in situ, or invasive SCC. Some OSCCs may manifest as ulcers with or without adjacent white or red mucosal changes. When invasion occurs, mucosa exhibits an increasingly irregular, granular, and ulcerated appearance. Continued growth leads to an exophytic or endophytic mass with a raised, rolled border [1,4].

Dental practitioners play a significant role in diagnosing oral cancer compared with family physicians since a conventional oral examination can efficiently detect oral cancer [5,6]. Therefore, it is important that dental practitioners examine all of their patients. In particular, patients at high risk for oral cancer should be thoroughly examined. All dental students should also be familiar with and aware of the early signs of oral cancer. If potentially malignant disorders are documented in their earliest stages, malignant changes can possibly be prevented, and the success rates for therapy will be higher [7]. In addition to a better prognosis, early detection of oral cancer will have a lower cost of treatment than that for advanced stages of oral cancer. Furthermore, multiple studies have shown that early diagnosis of small localized lesions is associated with a survival rate of 70–90% [8,9,10,11].

Clinicians should recognize that any lesion exhibiting similarities to ulcers, including red or white plaques, which remains in the mouth for more than two weeks, may be an indication of malignancy [12]. However, dental professionals do not always examine the mouth properly, and a large number of cases are not discovered. This can be due to the dentist’s poor practice, non-specific clinical appearance of the lesions, lack of oral cancer knowledge, and/or delays in referrals and treatment. In addition, some professionals take a biopsy from an incorrect spot, which leads to a wrong diagnosis and a delay in confirming disease [13,14]. When a general dentist performs oral cavity screening, it should also include a thorough medical/dental history and physical examination [15].

In the present study, the aim was to assess the diagnostic accuracy of future dental practitioners when presented with mucosal changes suggestive of oral cancer and potentially malignant disorders in a survey-based questionnaire. An early diagnosis of oral cancer can minimize disease mortality, improve patient quality of life, and prevent complications and disfigurement [16]. Thus, it is essential for dental practitioners to be able to detect and recognize different types of oral lesions.

## 2. Materials and Methods

For this study, a cross-sectional questionnaire-based survey was conducted at the College of Dentistry, Ajman University (AU). This study received ethical approval from the Research Ethics Committee (REC) of AU (Ref: D-F-H-19-03-04), and all participants completed an informed consent prior to participation. The questionnaire forms were electronically distributed to fifth-year students, interns, and recently graduated General Dental Practitioners (GDPs) who had completed their internship within the previous three years. The questionnaire included 32 case scenarios of patient and lesion histories accompanied by high resolution images of a broad range of representative oral lesions (Figure 1). The cases included in the questionnaire were selected by a panel of oral medicine specialists to represent the following categories: (1) normal variation of anatomy of the oral cavity, (2) benign oral mucosal lesions, (3) oral potentially malignant disorders (OPMDs), and (4) oral malignancies. For each case, the participants were requested to make a provisional diagnosis of the case based on evaluating the provided history and clinical photograph and to determine the potential of malignancy by choosing one of the following options: (a) Unlikely, (b) Likely, (c) Definitive potentially malignant, (d) Frank malignant.

Based on these responses, the respondent’s ability to diagnose oral lesions of varying malignant potential and their knowledge of oral cancer appearance were assessed. Each participant was also asked to provide demographic data (gender, and level of education). Given that the current study was not designed as an in vivo diagnostic test accuracy study, we used appropriately some elements of the Standard for Reporting Diagnostic Accuracy (STARD) tool.

Data were entered and analyzed in IBM SPSS 22 (IBM Corp, Armonk, New York, NY, USA). Mean (%), valid mean (%) for different criteria and tests (e.g., normality test), hypothesis test summary, independent-sample test summary, and Mann-Whitney U-test summary data were compared. The difference in the knowledge score test was assessed using a cluster-adjusted analysis by considering the intra-class correlation coefficient.

The study has followed a convenience sampling method as the investigators invited all eligible subjects (total number = 211) in the institute to participate in the study.

## 3. Results

### 3.1. Overall Results

A total of 211 eligible participants were invited to respond to a survey-based questionnaire. These were divided as following: fifth-year dental students (*n* = 99), dental interns (*n* = 67), and general dentists (*n* = 45). Only 132 valid surveys were returned, resulting in a response rate of 62.5%. Among these 132 respondents, 84 (63.6%) were female and 48 (36.4%) were male. The respondents included fifth-year dental students (*n* = 80), dental interns (*n* = 39), and GDPs (*n* = 13).

The overall mean rate of correct answers was 46%. According to the four categories of cases included in the questionnaire (normal variation, benign lesions, malignant tumor, and OPMDs), the percentage of correct answers was 60.3%, 31.0%, 55.7%, and 33.4%, respectively.

### 3.2. Diagnostic Accuracy According to Gender

When overall diagnostic accuracy was evaluated according to participants’ gender, there was no statistically significant difference (females, 48.3%; males, 47.2%). When the percentage of correct answers for each of the four categories for male versus female respondents was analyzed, there were no significant differences (p, 0.231) (Table 1). However, it was observed that both male and female participants exhibited difficulty in identifying benign and premalignant lesions. For example, the rates of correct answers were <35% for these two categories compared with > 55% for normal variations and malignant tumor cases.

### 3.3. Diagnostic Accuracy According to Education Level

When we compared diagnosis accuracy according to education level, the highest percentage of correct answers was achieved by the interns (52.5%), followed by the GDPs (51.9%) and fifth-year students (44.1%). Mann Whitney test showed that a statistical significant difference was found between two groups; interns and fifth-year students (p, 0.016). However, according to category (normal variation, benign tumor, malignant tumor, and premalignant lesion), the GDPs returned the highest percentage of correct answers for three out of the four categories (Table 2). Only for malignant tumor cases did interns have a higher percentage of correct answers.

### 3.4. Knowledge Scores

Knowledge scores were calculated for the three groups of respondents. They were: 30.78 ± 7.16 for the fifth-year students, 34.23 ± 5.5 for the interns, and 33.46 ± 6.86 for the GDPs. For GDPs, an obvious range difference was also observed, while the range for the interns was less.

The mean knowledge scores for females versus males were 32.08 ± 6.18 and 32.04 ± 7.9, respectively. For both male and female participants, the highest rates of correct answers were associated with normal variation of anatomy of the oral cavity cases, followed by malignant tumors, OPMDs, and benign lesions.

## 4. Discussion

To improve the survival rate of patients with oral cancer malignancies, early detection and diagnosis of oral potentially malignant disorders is critical. An inability to recognize mucosal changes and lesions and the extent of their seriousness can lead to major health issues for patients, and in some cases, death. In this study, a 32-question validated questionnaire survey was used to assess whether a cohort of fifth-year dental students, interns, and GDPs were able to detect clinical features suggestive of oral malignancies.

The questionnaire responses did not significantly differ according to gender, consistent with previous results [17,18,19].

While the overall percentage of correct answers in the present study was 46%, consistent with previous studies that have sought the opinions and attitudes of dental students and professionals on their knowledge and practices on oral cancer [6,19,20], this result indicates that greater education and individual knowledge of oral cancer should be achieved and maintained. The unique design of our survey is that participants were asked to review a case history and representative clinical photographs to follow a real scenario of daily practice. Normal variations of anatomy of the oral cavity and malignant lesions were most often diagnosed correctly, followed by OPMDs and benign lesions. Greater recognition of malignant lesions is consistent with the more distinct features of these lesions. Conversely, the lowest percentage of accurate answers for benign lesions is consistent with their similarities to normal variations. Cases involving normal variations were correctly identified >50% of the time. The latter result may be due to the repetition of information during dental course and in different study modules.

Several studies evaluated the knowledge, views, and management regarding oral cancer of dental students and dentists in the United Kingdom [6,21,22], United States [19,23], Canada [24], Europe [25], Saudi Arabia [5,26], and UAE [27]. All of these studies came to the conclusion that knowledge regarding preventing and identifying oral cancer in the dental field needs to be improved. Our study findings have echoed that and presented a robust methodology to assess the intended learning outcomes of teaching oral cancer and OPMDs modules in a dental curriculum.

Regarding the level of education and related accuracy of diagnosis, a statistically significant difference in the mean knowledge scores between the interns and fifth-year students was observed (p 0.016), with the interns having a higher mean score (30.78 ± 7.16 and 34.23 ± 5.5, respectively). The scores between the interns and GDPs were also very close (34.23 ± 5.5 and 33.46 ± 6.86, respectively). When we examined the percentage of correct diagnoses for the cases involving normal variations, benign lesions, malignant tumors, and OPMDs, we observed a similar pattern among the fifth-year students, interns, and GDPs. The percentage of correct diagnosis was highest for cases of normal variations, followed by cases of malignant tumors, OPMDs, and benign lesions. Furthermore, the GDPs had the highest percentage of correct answers in all of the categories except malignant tumors (which the interns had a higher score). Conversely, in all four categories of lesion identification, the fifth-year students had the lowest percentage of correct answers. Interestingly, students in their final year of dental study exhibited greater awareness and knowledge of oral cancer than those in their lower grades [28]. In another study that compared the knowledge of students at different levels, the mean knowledge score was found to positively correlate with greater years of education [20]. Characteristics of fifth-year students have been found to include greater concern for improving their hand skills during clinical work rather than for course requirements and focused attention on dental characteristics of the oral cavity rather than more extensive considerations of soft and hard tissues when checking for abnormalities [20]. Fifth-year students also have less experience than GDPs or interns since they have not been exposed to as many cases, and they do not have a lot of free time to attend lectures or campaigns outside of university obligations.

This study demonstrates that despite meeting the educational objectives of the dental course in the institute, translating the knowledge on oral cancer and OPMDs into effective diagnostic ability is still a challenge. Efforts to set a comprehensive and universal modules to learn the clinical skills of diagnosing mucosal changes suggestive of oral malignancy or premalignancy is a paramount to improve the training of oral oncology.

Despite the fact that the study has limitations due to the small sample size and being conducted in one geographic region, the relevance of the present results is of great note in the Middle East where the incidence of oral cancer is increasing [29,30]. Another limitation lies in the use of photographs where participants can’t palpate the lesions. However, we provided a case scenario outlining the major risk factors and lesion characteristics to simulate the clinical environment.

## 5. Conclusions

In conclusion, the responses highlight difficulties in discovering and identifying different types of lesions in oral mucosa by the future dentists. Therefore, reinforcement of clinical skills for current and upcoming dental professionals for the detection of high-risk patients and etiological factors causing oral cancer are greatly needed. For students, oral cancer inspection and management guidelines need to be reinforced. Increased emphasis in teaching curriculum on different types of pathological oral lesions would also help, as well as examinations of clinical oral cases. Further consideration of the continuous professional development courses regarding oral cancer and OPMDs should be undertaken to improve the diagnostic skills of practicing dentists.

## Figures and Tables

**Figure 1 healthcare-09-00263-f001:**
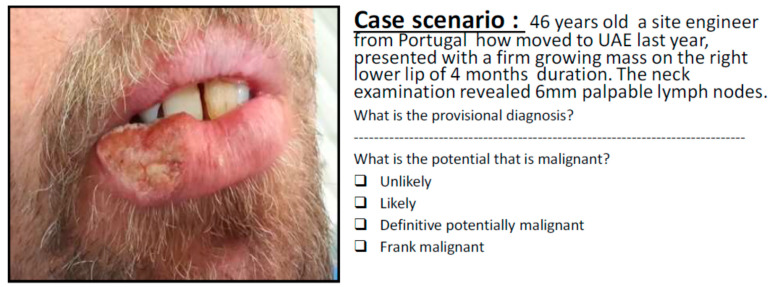
A case scenario representative of malignancy arising on the lower lip.

**Table 1 healthcare-09-00263-t001:** Percentage of correct answers per category according to gender.

Category	Female	Male
Normal variation	67.3%	66.4%
Benign lesion	32.3%	28.8%
Malignant tumor	55.4%	58.7%
Oral potentially malignant disorder	34.5%	31.6%

**Table 2 healthcare-09-00263-t002:** Percentage of correct answers per category according to education level.

Level of Education	Normal Variation	Benign Lesion	Malignant Tumor	Oral Potentially Malignant Disorder
Fifth-year	63.25%	27.00%	49.75%	32.24%
Intern	71.72%	30.23%	68.21%	34.73%
GDPs	75.00%	32.58%	64.62%	35.00%

## Data Availability

The data that support the findings of this study are available on reasonable request from the corresponding author. The data are not publicly available due to privacy or ethical restrictions.

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
