# Peer review of "Diagnostic Accuracy of Oral Cancer and Suspicious Malignant Mucosal Changes among Future Dentists"

_healthcare, 2021, doi:10.3390/healthcare9030263_

Round 1
Reviewer 1 Report
Early oral cancer detection is of paramount importance in dental practice but clearly it cannot happen if practitioners are not educated on the matter. I believe the information provided are interesting, the article is also clear, brief, and well-written, although this article is similar to other papers that have already researched this argument. I believe some small changes that could overall improve the article should be conducted.
- If the authors received an approval code from the ethical committee, they should include it in the paper, as only stating that they received the approval is not sufficient.
- I believe that inserting a couple clinical pictures, and some examples of the clinical histories chosen from the ones that were inserted in the questionnaire might overall improve the article, as it would explain the classification adopted and make it more interesting.
- The questionnaire was only mentioned, but I think the authors should better explain how the question were posed to the interviewed, and therefore provide a practical example of the questionnaire.
- The statistical part of the MM section should be improved, providing the reader with some more information, especially on the adopted knowledge test, as it is not clear hot the reported methods were used.
- The authors should clarify how the sample was selected, as this might have an influence on the results, and they should report on the qualification of the 211 initial participants as they did of the 132 subjects that responded the questionnaire.
- The Result section is clear and well laid out, although it is not clear where the Mann-Whitney U-test included in the MM section was adopted.
- P-values of statistical results should be included in the Results section.
- “The questionnaire responses did not significantly differ according to gender, consistent with previous results” This phrase should be reworded as the so-called previous results all come from cohort different from the one included in this study, and the methods adopted in the cited studies are different from the one used in this paper.
- “Similar results were obtained in a study conducted in Iran” The cited study had also a different study group as only students were enrolled and had a different investigation method.
- Greater awareness, campaigning, and the wider availability of techniques for early detection of cancerous lesions and potentially malignant oral diseases is relevant in all regions where oral cancers are diagnosed [27, 31]. This phrase could be reworded as it is merely stating the obvious.
Author Response
We thank the reviewer for their time and efforts to improve the clarity of our manuscript. We have revised the manuscript per the provided comments.
Early oral cancer detection is of paramount importance in dental practice but clearly it cannot happen if practitioners are not educated on the matter. I believe the information provided are interesting, the article is also clear, brief, and well-written, although this article is similar to other papers that have already researched this argument. I believe some small changes that could overall improve the article should be conducted.
- If the authors received an approval code from the ethical committee, they should include it in the paper, as only stating that they received the approval is not sufficient.
Authors response: reference number was added.
- I believe that inserting a couple clinical pictures, and some examples of the clinical histories chosen from the ones that were inserted in the questionnaire might overall improve the article, as it would explain the classification adopted and make it more interesting.
Authors response: The full survey was appended as a supplementary material. We also added figure 1 to illustrate one case scenario.
- The questionnaire was only mentioned, but I think the authors should better explain how the question were posed to the interviewed, and therefore provide a practical example of the questionnaire.
Authors response: Additional details on the questionnaire structure were added.
- The statistical part of the MM section should be improved, providing the reader with some more information, especially on the adopted knowledge test, as it is not clear hot the reported methods were used.
Authors response: The statistical section has been expanded to provide additional information.
- The authors should clarify how the sample was selected, as this might have an influence on the results, and they should report on the qualification of the 211 initial participants as they did of the 132 subjects that responded the questionnaire.
Authors response: A statement on the sample size was added to the methods section, and the total number of eligible participants was outlined per subgrouping.
- The Result section is clear and well laid out, although it is not clear where the Mann-Whitney U-test included in the MM section was adopted.
Authors response: This has been clarified.
- P-values of statistical results should be included in the Results section.
Authors response: This has been added as needed.
- “The questionnaire responses did not significantly differ according to gender, consistent with previous results” This phrase should be reworded as the so-called previous results all come from cohort different from the one included in this study, and the methods adopted in the cited studies are different from the one used in this paper.
Authors response: Although the cited studies have used different types of questionnaires, but the gender has shown to be non-significant factor in making difference in the participants response. We believe this benchmark is valid.
- “Similar results were obtained in a study conducted in Iran” The cited study had also a different study group as only students were enrolled and had a different investigation method.
Authors response: Valid point. The statement has been amended.
- Greater awareness, campaigning, and the wider availability of techniques for early detection of cancerous lesions and potentially malignant oral diseases is relevant in all regions where oral cancers are diagnosed [27, 31]. This phrase could be reworded as it is merely stating the obvious.
Authors response: The statement has been deleted because of redundancy.
Reviewer 2 Report
Thank you very much for the precious opportunity. I have read it carefully.
Abstracts
Well organized, but no description of statistical analysis results. It is stated that there is no significant difference, but the lack of significant difference is also an important result and needs to be stated.
introduction
The background is well written.
Methods
Please include the approval number of the Ethics Review Committee.
Result
The results section describes the statistical analysis methods (last column on page 3). This should be listed in the Statistical Methods section of the Methods.
The results of the Knowledge scores should also be tabulated and all results should be listed.
All results should be listed, even if they are not significant.
Please include the actual P value, not the P<0.05 value.
Disscussion
Do not repeat the results in the Discussion section.
It is inappropriate to compare the results of different evaluation methods and assume that they are equivalent. You should carefully consider each country's paper individually.
In addition, it is presumed that the data is negative with almost no significant difference. I think it is inappropriate to assume that each country's data is equivalent. Please only discuss what you have learned from the results.
All other information should be included in the Future study section.
All in all, unfortunately, there is no new knowledge or data suitable for improving educational methods, so we consider this to be a study of low novelty.
Author Response
Thanks for your time and efforts in providing your feedback. We have revised our manuscript per your comments.
Here is our response to your comments.
Thank you very much for the precious opportunity. I have read it carefully.
Abstracts
Well organized, but no description of statistical analysis results. It is stated that there is no significant difference, but the lack of significant difference is also an important result and needs to be stated.
Authors response: the abstract has been modified accordingly.
introduction
The background is well written.
Authors response: many thanks for the kind words
Methods
Please include the approval number of the Ethics Review Committee.
Authors response: This has been added.
Result
The results section describes the statistical analysis methods (last column on page 3). This should be listed in the Statistical Methods section of the Methods.
The results of the Knowledge scores should also be tabulated and all results should be listed.
All results should be listed, even if they are not significant.
Please include the actual P value, not the P<0.05 value.
Authors response: The results section has been amended accordingly.
Disscussion
Do not repeat the results in the Discussion section.
It is inappropriate to compare the results of different evaluation methods and assume that they are equivalent. You should carefully consider each country's paper individually.
In addition, it is presumed that the data is negative with almost no significant difference. I think it is inappropriate to assume that each country's data is equivalent. Please only discuss what you have learned from the results.
All other information should be included in the Future study section.
Authors response: valid comments. The discussion section has been amended and the cited studies were discussed as relevant.
All in all, unfortunately, there is no new knowledge or data suitable for improving educational methods, so we consider this to be a study of low novelty.
Authors response: The current data indicate a need for additional teaching hours of oral cancer and OPMDs in the dental curriculum. It reiterates the message that several health and dental education bodies need to focus on this issue.
Reviewer 3 Report
I congratulate to the Authors for the idea of the study and the used methods.
Oral cancer rate is still to high and prevention camapins and awareness among clinicians, patients and national healthcare systems are strongly necessary and to be improved.
I only suggest to provide a copy of the questionnaire so to examine the quality of the pictures.
Author Response
We thanks the reviewer for your kind comments and we have supplied the questionnaire as a supplementary material
Round 2
Reviewer 2 Report
You have not properly added the necessary information on the matters I have pointed out. Also, the research question of this study is unknown as before. The methodological research design and statistical methods are also inappropriate. Therefore, there is no need to discuss this study any further.
Thank you very much.
Author Response
We thank the reviewer for the time and effort in reviewing our manuscript.
We revisited our previous revision regarding the points that you mentioned previously and have made a new revision to satisfy your comments.
Our study findings have echoed the results from previous papers. However, it presents a robust and new methodology to assess the intended learning outcomes of teaching oral cancer and OPMDs modules in a dental curriculum.
Assessing the competency of future dentists will enable to improve the relatively poor survival rates of oral cancer.